# Characteristics of Social Media Content and Their Effects on Restaurant Patrons

**June-Hyuk Kwon** [1] , **Sally Kim** [2], **Yong-Ki Lee** [3,*] **and Kisang Ryu** [4]

1 Graduate School of Business, Sejong Cyber University, Seoul 05000, Korea; kjh_business@naver.com
2 School of Business, Shenandoah University, Winchester, VA 22601, USA; ykim2@su.edu
3 School of Business, Sejong University, Seoul 05006, Korea
4 College of Hospitality and Tourism Management, Sejong University, Seoul 05000, Korea; kryu11@sejong.ac.kr
* Correspondence: yongki2@sejong.ac.kr; Tel.: +82-2-3408-3158

**Abstract:** The purpose of this study is to examine four characteristics of social media content and their effects on restaurant patrons. The characteristics we examine in our study are authenticity, consensus, usefulness, and aesthetics. More specifically, the study investigates how content characteristics influence consumers' cognition-based and affect-based attitudes toward the message, which, in turn, influence brand attitude and behavioral intentions. Data were collected from 376 respondents who had frequented restaurants on a regular basis and used social media networks for at least one year. Structural equation modeling with AMOS 22.0 was used to analyze the data. The most important content characteristic that influences both cognition-based attitude and affect-based attitude is usefulness. All three other content characteristics (authenticity, consensus, and aesthetics) are also found to have a significant impact on either cognition-based or affect-based attitudes. While both cognition-based and affect-based attitudes have a significant effect on brand attitude, the effect of affect-based attitude is fully mediated by brand attitude in influencing behavioral intentions. The mediating role of brand attitude is also examined.

**Keywords:** social media content; attitude; behavioral intentions; authenticity; consensus; usefulness

## 1. Introduction

The number of social media users has grown exponentially, making firms shift their focus from traditional communication channels (e.g., TV) to digital channels, particularly social media. One of the most important and distinguished characteristics of social media is its interactivity. Unlike traditional communication, in which firms push one-way communication to the recipients, social media allows consumers to share and exchange information, opinion, and knowledge [1]. Social media's inherent nature of interactivity means information can be spread rather quickly. Some social media features such as "Likes" on Facebook and "retweet" on Twitter also help with information dispersion. Another noteworthy characteristic of social media is concerned with perceived credibility. Research shows that consumer-posted information is perceived as more credible than firm-posted information [2]. For example, consumers tend to believe information posted by fellow consumers more than an advertising message claimed by the company. While the interactive nature of social media helps expedite information dispersion, and, thus, is related with the quantity of people the message reaches, credibility has something to do with the perceived quality of the message. These two characteristics constitute the magnitude of impact of social media on consumers. Social media is the most influential marketing communication tool that 63 percent of the restaurants used in 2018 (https://www.modernrestaurantmanagement.com/10-social-media-marketing-tips-for-restaurants/). Social media enhances and cultivates a long-term relationship with customers [3] and plays a critical role in increasing restaurant performance [4]. This means that social media can serve as a platform to generate consumer-led content [5], and should be considered one of the strategic tools for

businesses to create positive attitude [6]. Therefore, restaurant businesses dealing with social media should think about how to manage their brand in the realm of social media. In response to the call for theoretical explanations of the role of social media platforms in creating positive attitude and enhancing customer loyalty in the restaurant context, the purpose of the current study is to identify characteristics of social media content that are relevant to restaurant patrons. Studying the role of social media in the restaurant industry is worthwhile because restaurant patrons heavily rely on social media as an essential tool for gathering information and making a purchase decision. Therefore, our study aims to examine four characteristics of social media content that influence restaurant patrons. The second research objective is concerned with the mechanism through which content characteristics influence consumers. Based on the well-known Theory of Reasoned Action (TRA) [7], Elaboration Likelihood Model (ELM), and social influence theory [8], we examine the effect of content characteristics on attitude to the message (cognition-based attitude and affect-based attitude), brand attitude, and behavioral intentions. Although some research claims that social media influences consumer responses [3] and business performance [4], we are not clear about the underlying mechanism by which consumers respond in the restaurant context.

There are several contributions of our study. The first contribution is related to its examination of four content characteristics pertinent to social media in the restaurant industry. While some previous studies [3,9,10] examined social media content characteristics, they were either very general (e.g., social media website quality [10]) or narrowly focused on one element (e.g., web aesthetics [11]). Our study examining four dimensions enhances our understanding of the distinct impacts of content characteristics in the restaurant industry. The finding will help restaurant managers identify social media content characteristics that maximize consumer responses, and allocate appropriate resources to maximize the return on their communication efforts. Second, our study makes a contribution to the literature by adopting and testing a dual system of attitude (cognition-based attitude and affect-based attitude). There are some studies that investigated a relationship between marketing communication input and attitude toward the message, but very few studies considered both cognition-based and affect-based attitudes [12]. Examining two types of attitude to the message is critical for making a connection between content characteristics and consumer responses, based upon which we can draw significant theoretical and strategic implications. Another contribution of our study is related to its treatment of brand attitude as a mediator between attitude to the message and behavioral intentions. Since our study employs a dual system of attitude, we can identify a distinctive role brand attitude plays in mediating the effect of cognition-based and affect-based attitudes on behavioral intentions. Lastly, our study, grounded on the ELM, adds further evidence to the body of knowledge on how consumers process information in the context of social media.

## 2. Theoretical Framework

### 2.1. Marketing Communication Process and Attitude

Researchers have used different theories and frameworks (e.g., TRA, social judgment theory) to explain how marketing communication works. One of the common elements shared among various frameworks is stimulus-response, which denotes that marketing communication is a stimulus intended to provoke a response in the recipients. The goal of marketing communication is to influence message recipients (consumers) so that they form a favorable attitude toward the message and, subsequently, the product/brand. Attitude is typically referred to as enduring and general positive or negative evaluation about an object, person, or thing [13]. There is no single conceptualization about the term attitude. While some researchers view attitude as being drawn from cognitive judgment (e.g., belief about a brand) [14,15], and, thus, relate attitude to belief and knowledge, others consider feelings in defining attitude [13]. Our study views attitude as being comprised of both cognitive and affective elements.

Fishbein and Arjen's [7] TRA is helpful for understanding how marketing communication influences consumers in terms of attitude and behaviors. Fishbein and Arjen [7] theorize that people, upon evaluating attributes of an object, form an attitude, and the attitude affects behavioral intentions. Based on the TRA, we view that restaurant patrons will evaluate a social media message and form an attitude toward the message, which influences their attitude toward the restaurant. We also examine a link between attitude and behavioral intentions. Simply put, our framework considers four elements: social media content characteristics, attitude toward message, attitude toward brand, and behavioral intentions. Finally, Kelman's [8] social influence theory is helpful for understanding how consumers evaluate characteristics of social media. The theory assumes that people in social networks are influenced by others as they try to conform to others' behavioral patterns. It prescribes the process in which consumers' learning occurs as a result of the influence of different sources of information [16].

Our model has a couple of merits. First, unlike some studies [17,18] that examined a direct link between marketing communication input and purchase measures (e.g., sales, purchase intentions), our model considers intermediate constructs (i.e., attitude toward the message, attitude toward the brand), through which the effect of marketing communication is linked to behavioral measures. Thus, our study treats consumer's response to a message as a mediator and intends to examine the mediating effect in the marketing communication process. Our approach is consistent with many of the previous studies that considered intermediate constructs [19–22]. For example, in their study of a meta-analysis, Brown and Stayman [20] show a significant effect of consumer's attitude to marketing communication on brand attitude, which, in turn, influences purchase intentions. Similarly, Mitchell and Olson [22] show that attitude to marketing messages serves as a mediator between marketing communication input and brand attitude.

Second, our model captures both cognitive and affective responses to a marketing communication input, integrating two important dimensions. Some previous studies included an intermediate construct in their model, but focused on either cognitive or affective response [19,21,23,24]. Earlier studies [25,26] focused on cognitive responses (e.g., belief, knowledge). The concept of affect started to gain significant attention from researchers in 1980s, proliferating studies on the role of affect in marketing communication [19,27]. For example, Batra and Ray [19] examined consumer's affective response to advertisement and showed a significant relationship among affective response, attitude to advertisement, attitude to brand, and purchase intentions. Similarly, Burke and Edell [28] showed feelings (affective response) associated with a marketing communication message influenced consumer's attitude toward the message. In addition, some researchers [23,24] emphasizing the role of affect suggested that visual and emotional elements are more important than information on the product. These studies made an important contribution to the literature by delving into the role of affect in attitude formation. However, in doing so, some failed to consider cognition in their model [19,27]. Some researchers [12,27,29] stress the importance of considering both affective and cognitive responses in a communication model. For instance, Lee et al. [12] report that both cognitive and affective systems make a contribution to decision making and they operate differently leading to distinct outcomes. Therefore, our model includes both cognitive and affective responses and treats them as mediators between marketing communication input (i.e., social media message characteristics) and consumer's attitude toward the brand.

### 2.2. Social Influence and Conformity

Following prior research [30,31], our study based on the Kelman's [8] social influence theory, attempts to explain how social media messages influence restaurant patrons. According to Kelman 98, there are three distinct social influence methods: Compliance, identification, and internalization. Compliance is the acceptance of the influence for the purpose of seeking rewards or avoiding punishments (e.g., not changing an airline ticket so as to avoid a fee) [31]. Identification occurs when consumers adopt standards or behaviors

of the influencer or communicator because they desire affiliation. Prior research [32] on branding and celebrity endorsement used this concept to explain why consumers adopt a particular brand. For example, a teenager may choose a celebrity-endorsed brand because he/she wants to be identified with the image of the celebrity. Internalization, which is most relevant to our study, captures the degree to which consumers adopt the influence or behaviors because their value system is consistent with the influencer's or the communicator's. Marketing communication is about disseminating information with an objective of exerting an influence on the message recipients through internationalization of the information.

Lascu and Zinkhan [31] who conducted a review on consumer conformity studies propose a model in which consumer conformity is presented in two major manifestations: normative and informational influences. They categorize Kelman's (1958) compliance and identification as normative influences and internalization as informational influence. Lascu and Zinkhan [31] view that compliance and identification occur as a result of accepting the norm that is deemed appropriate by the influencer or the communicator. On the other hand, internalization takes place when consumers accept information as it is consistent with their existing value system, and, thus, is referred to as informational influence [31,33]. Grounded on Lascu and Zinkhan's [31] conformity framework, our study maintains that consumer's acceptance of a social media message occurs as a result of normative and informational influences. For example, a consensus shown on social media postings (e.g., many others chose a certain product) may move consumers to follow the majority's opinion, capitulating to the power of normative influence. We view social media messages influence consumers through normative and informational influences.

### 2.3. Characteristics of Social Media Message Content

Prior research [3,34–41] examines different characteristics of marketing communication content including vividness, usefulness, novelty, neutrality, reliability, authenticity, and consistency. Our study focuses on four major content characteristics that are pertinent to the restaurant industry. Restaurant customers tend to rely on others' opinions when gathering information and choosing a particular restaurant [42]. Prior research suggests that information should be authentic, useful, of consensus, and be presented aesthetically in order to be effective [10,12,21,41,43].

#### 2.3.1. Authenticity

We define authenticity as the degree to which a message is perceived to be genuine and trustworthy. Efforts have been made to specify what constitutes authenticity. Some [44–46] argue authenticity is drawn from historical facts, properties, and tradition, tying authenticity to objectivity. Others [47] maintain that authenticity is not only identified by objective properties but also a subjective perception of self or object. We adopt the latter approach because our study focuses on consumer's perception of authenticity, which is subjective. The literature [21,48–50] shows that consumers seek authenticity in goods, service experiences, leaders, celebrities, and marketing communication. Authenticity in marketing communication matters to consumers as they want to rely on the credible information to make an appropriate purchase decision. Kawalczyk and Pounders [21] examined celebrities and reported that a social media message perceived as authentic has a significant influence on the fans' emotional attachment with the celebrity and intentions to engage in positive word-of-mouth. Similarly, several studies [51–53] show that users' perceptions of message authenticity in social media have a positive effect on acceptance of the information.

#### 2.3.2. Consensus

Consensus, referred to as the majority opinion, has something to do with others' viewpoint [54] and functions as a heuristic cue about the object (e.g., brand) under consideration. Consensus refers to the percentage of the recipients agreeing on a specific opinion or view and it influences others' attitude [55]. Consensus plays a critical role in persuading the recipients by reducing perceptions of uncertainty [55].

Marketing researchers [9,17] recently started to examine and apply the concept of consensus to marketing communication and understand its role in persuading and influencing others. Several studies suggest that consumers use consensus as a diagnostic informational cue to make product judgments [9,17,56]. For example, Chang (2012) shows that females' intentions to purchase a product increase when they are exposed to ads that show consensus. Similarly, Benedicktus et al. [9] show that consensus information has a significant effect on consumers' trust and intentions to purchase. The effect of consensus seems to be manifested through its normative influence on the message recipients (consumers) as they may feel adopting the majority's viewpoint or behavior is expected or has some merits [31]. The effect of consensus may be more prominent in an online environment where consumers are engaged with sorting out, processing, and evaluating a vast amount of information. In an information overloaded situation, consumers may look for a heuristic cue such as consensus to have an efficient information search and evaluation. Research [57,58] shows that a message showing a strong consensus has a greater effect on message recipients.

### 2.3.3. Usefulness

We borrow one of the well-known models, Technology Acceptance Model (TAM) [19,39], to argue that consumers consider usefulness in their evaluation of social media messages. TAM has been tested and validated empirically in many fields. It suggests two elements that are helpful for expediting the acceptance of information technology or systems: usefulness and ease of use [18,59]. Based on the TAM, usefulness is one's perception of the information technology being useful for improving his/her job performance [37]. In consideration of the context of our study, we define usefulness as the degree to which consumers believe the content is useful for satisfying their information need. Han [60] shows that useful information posted on social media has a positive impact on consumer's trust and purchase intentions. Kwok and Yu [61] found that restaurant posts that provided information, such as restaurant menus, were the ones that got more likes and comments. They also found that more straightforward messages such as photo and status update messages receive more attention (or reaction) from Facebook users than those containing a link or video. Similarly, other researchers [18,38,39] suggest that useful online information has a significant positive effect on consumer's attitude toward the brand and purchase intentions.

### 2.3.4. Aesthetics

Aesthetics refers to the degree to which one perceives a particular object as visually beautiful or pleasing [62]. Aesthetics has been studied in many different contexts including product design, retail environment, service experience, and marketing communication [11,41,63]. Prior research [63] supports that aesthetics of a physical environment (e.g., restaurant) has a significant influence on consumer's evaluation of the service experience. Recently, researchers started to pay attention to web aesthetics [11,41]. They suggest that websites not only fulfill utilitarian purposes of delivering information but also fulfill consumers' hedonic needs by offering an entertaining experience [64]. Wang et al. [11] show that web aesthetics have a positive impact on consumer's purchase intentions and search activity. Their study suggests that an aesthetic stimulus increases consumer's tendency to explore more on the website and purchase intentions. Similarly, other studies [65,66] display that aesthetics are an important factor especially in the service industry.

## 3. Hypotheses

### 3.1. The Impact of Social Media Content Characteristics on Attitude toward the Message

The ELM of persuasion proposed by Petty and Cacioppo [67] is a theoretical foundation based on which our study establishes a relationship between social media content characteristics and consumer attitude. Many studies in advertising and marketing communication have validated this model [68,69]. The ELM posits that elaboration of information takes two different routes: central route and peripheral route. The model dictates that the way two routes operate is a function of the degree of elaboration. Central route is used

for attitude formation or change when consumers engage in careful and close scrutiny of message-relevant arguments, requiring a high level of elaboration. On the other hand, peripheral route is used when consumers rely on the elements that are not related to message arguments (e.g., attractiveness of the message source) but the peripheral cues evoking inferences about the message. A likely outcome is that the peripheral route requires less consumer involvement in the processing of information.

Built on the ELM, our study postulates that consumers use both central and peripheral routes in forming an attitude toward a message [70]. Under the central route, consumers will engage in careful consideration of true merits of the information and scrutinize the content. Thus, taking a central route is concerned with evaluating quality of the content or the message argument. Authenticity and usefulness of the message are related with quality of the content, which require a high level of involvement, analytical processing, and elaboration. On the other hand, a peripheral route of processing is likely to be taken when consumers rely on peripheral cues such as consensus and aesthetics. Consumers using the peripheral route will rely on heuristic cues (e.g., aesthetics, consensus) in forming an attitude toward a message. According to the ELM and prior research, both central and peripheral routes will lead to attitude change [13,70]. In sum, our study anticipates that all four characteristics of social media content (either through the central route or the peripheral route) will have a positive influence on consumer's attitude toward the message (both cognitive and affective responses). Therefore, we hypothesize the following:

**Hypothesize 1 (H1).** *Authenticity has a positive effect on consumer's cognition-based attitude and affect-based attitude toward the message.*

**Hypothesize 2 (H2).** *Consensus has a positive effect on consumer's cognition-based attitude and affect-based attitude toward the message.*

**Hypothesize 3 (H3).** *Usefulness has a positive effect on consumer's cognition-based attitude and affect-based attitude toward the message.*

**Hypothesize 4 (H4).** *Aesthetics has a positive effect on consumer's cognition-based attitude and affect-based attitude toward the message.*

*3.2. The Influence of Attitude toward the Message on Brand Attitude and Behavioral Intentions*

In discussing the influence of attitude toward the message on brand attitude and behavioral intentions, we use a well-established hierarchy response framework. The framework specifies that behavior is predicted by attitude. Many previous studies [19,20,22,70] considered both attitude toward the message and attitude toward the brand in their models to understand the relationship between the two constructs. Lord et al. [70] show that attitude toward the brand is predicted by attitude toward the message. Similarly, Mitchell and Olson [22] report that attitude toward the message mediates the relationship between advertisement and attitude toward the brand. Brown and Stayman [20] in their meta-analysis show a positive relationship between attitude toward the message and attitude toward the brand. In addition, Lee et al. [71,72], found that attitude toward an event can be transferred to attitude toward the hosting country, suggesting that attitude can be transferred when an association between the two objects evaluated (e.g., an event and the hosting country) is established. These previous studies suggest that attitude toward the message mediate the relationship between marketing communication input and brand attitude. Based on prior research, we expect that both cognition-based attitude and affect-based attitude toward the message will have a positive influence on brand attitude [19,22].

In addition to their significant impact on brand attitude, both cognition-based attitude and affect-based attitude toward the message are expected to have a direct impact on behavioral intentions to purchase and spread positive word-of-mouth. Based on the TRA, we expect that effective marketing communication will have an influence on consumer's attitude toward the message, which, in turn, affects behaviors. Prior research [70] supports

our expectation by showing a positive link between attitude toward the message and positive behavioral intentions (e.g., intentions to purchase).

**Hypothesize 5 (H5).** *Cognition-based attitude toward the message has a positive effect on brand attitude.*

**Hypothesize 6 (H6).** *Affect-based attitude toward the message has a positive effect on brand attitude.*

**Hypothesize 7 (H7).** *Cognition-based attitude toward the message has a positive effect on behavioral intentions.*

**Hypothesize 8 (H8).** *Affect-based attitude toward the message has a positive effect on behavioral intentions.*

### 3.3. The Influence of Brand Attitude on Behavioral Intentions

Based on the TRA, we argue that brand attitude affects behavioral intentions to purchase and spread word-of-mouth. Studies suggest that favorable brand attitude leads to behavioral intentions to purchase [19,70,73,74]. For example, Lord et al. [70] report a positive impact of brand attitude on purchase intentions. Since the relationship between brand attitude and behavioral intentions is well established, we do not devote a lot of space to the discussion of this well-established relationship.

**Hypothesize 9 (H9).** *Brand attitude has a positive effect on behavioral intentions.*

## 4. Methodology
### 4.1. Sample and Data Collection

Data were collected using an online survey on consumers living in S. Korea who had frequented restaurants at least four times a month and used a social media network (e.g., Facebook, Twitter, Instagram, Naver Band) for a minimum of one year. We hired an online research company for data collection. Using a convenience sampling method, the company collected data in March of 2018. The company sent out emails to 1000 consumers on the panel and asked them to participate in the study. Those who did not meet the sampling criteria (dining frequency of a minimum of 4 times a month, a minimum 1 year of using a social media network, and a minimum of 25 years in age) were excluded from the study. The final sample consisted of 376 respondents.

### 4.2. Measures

All items were measured on a 7-point Likert-type scale anchored by "strongly disagree" and "strongly agree" (see Table 1). The four characteristics of social media content were measured with fourteen items all together based on previous studies [71,72,75–77]. Coefficient alphas for authenticity, consensus, usefulness, and aesthetics were 0.905, 0.774, 0.851, and 0.845, respectively. Cognition-based attitude and affect-based attitude toward the message were measured with five items adopted from the studies of Hwang et al. [78] and Lee and Lim [79]. Coefficient alphas for Cognition-based attitude and affect-based attitude were 0.794 and 0.910, respectively. We used three items for measuring brand attitude based on prior research [19,22,80,81]. Coefficient alpha for brand attitude was 0.901. Finally, three behavioral items were used to measure behavioral intentions related to brand loyalty (i.e., intentions to purchase, intentions to spread word-of-mouth) adopted from the study of Chaudhuri and Holbrook [74]. Coefficient alpha for behavioral intentions was 0.893.

**Table 1.** Measurement model result from confirmatory factor analysis.

| Constructs and Items | Standardized Factor Loadings | Skewness | Kurtosis |
|---|:---:|:---:|:---:|
| Authenticity (CCR [1] = 0.849, AVE [2] = 0.533) | | | |
| The SNS restaurant content I've searched for or viewed is largely credible. | 0.872 | −0.291 | 0.289 |
| I thought that the SNS restaurant content I searched for or viewed was trustworthy. | 0.918 | −0.283 | 0.008 |
| The SNS restaurant content I've searched for or viewed is constant. | 0.797 | −0.110 | 0.341 |
| The SNS restaurant content I've searched for or viewed is pure. | 0.766 | −0.148 | 0.219 |
| The SNS restaurant content I have searched for or viewed is non-commercial. | 0.636 | −0.051 | 0.455 |
| Consensus (CCR = 0.798, AVE = 0.570) | | | |
| The number of SNS restaurant content that I searched for or saw was large. | 0.717 | −0.440 | 0.908 |
| The SNS restaurant content I searched for or saw was positive. | 0.676 | −0.339 | 1.248 |
| Many people agreed with the SNS restaurant content I searched for or viewed. | 0.792 | −0.224 | 1.135 |
| Usefulness (CCR = 0.814, AVE = 0.594) | | | |
| The SNS restaurant content that I have searched for or seen is useful in everyday life. | 0.806 | −0.253 | 0.488 |
| The SNS restaurant content I searched for or viewed provided useful information. | 0.833 | −0.320 | 0.726 |
| The SNS restaurant content I've searched for or viewed has allowed me to spend economically. | 0.767 | −0.442 | 0.603 |
| Aesthetics (CCR = 0.822, AVE = 0.607) | | | |
| The SNS restaurant content I've searched for or viewed has stimulated my appetite. | 0.795 | −0.193 | 0.498 |
| SNS restaurant content that I searched for or saw was abundant with aesthetic elements, such as menus and store interiors. | 0.828 | −0.072 | 0.075 |
| I think the SNS restaurant content I have searched for or viewed is pretty. | 0.794 | −0.073 | 0.052 |
| Cognition-based attitude toward message (CCR = 0.731, AVE = 0.773) | | | |
| I felt that the SNS restaurant content I encountered was unique. | 0.725 | −0.323 | 0.452 |
| I felt confident after viewing SNS restaurant content. | 0.886 | −0.425 | 0.442 |
| Affect-based attitude toward message (CCR = 0.865, AVE = 0.682) | | | |
| I felt happy after viewing SNS restaurant content. | 0.859 | −0.366 | 0.632 |
| I had fun viewing SNS restaurant content. | 0.866 | −0.578 | 0.678 |
| I felt that the SNS restaurant content was attractive. | 0.881 | −0.483 | 0.754 |
| Brand attitude (CCR = 0.861, AVE = 0.675) | | | |
| I became familiar with the brand after viewing SNS restaurant content. | 0.816 | −0.499 | 0.698 |
| I began liking the brand after viewing SNS restaurant content. | 0.875 | −0.498 | 1.109 |
| I became interested in the brand after viewing SNS restaurant content. | 0.839 | −0.470 | 1.279 |
| Behavioral intentions (CCR = 0.866, AVE = 0.683) | | | |
| We will continue to use the brand after receiving the SNS restaurant content. | 0.863 | −0.532 | 0.929 |
| I am willing to recommend the brand to others after encountering SNS content. | 0.845 | −0.459 | 0.772 |
| We will reuse the brand after encountering SNS restaurant content. | 0.863 | −0.473 | 1.022 |

$\chi^2$ = 404.548 (df = 244, *p* = 0.000, $\chi^2$/df = 1.658), GFI = 0.922, NFI = 0.945, CFI = 0.977, RMSEA = 0.042. [1] Composite construct reliability. [2] Average variance extracted.

## 5. Result

### 5.1. Demographic Profile

Table 2 presents a demographic profile of the sample. The sample consisted of 51% of females and 49% of males. More than half of the respondents (71%) were in the ages between 25 and 45. About 60% of the respondents visited social media network sites between four and nine times a month, followed by a group of respondents (28%) who visited between 10 and 15 times. More than half of the respondents (52%) used social media network sites for more than four years.

**Table 2.** Profile of the sample.

| Demographics | Frequency (n) | Percentage (%) |
|---|---|---|
| Gender | | |
| Male | 183 | 48.7 |
| Female | 193 | 51.3 |
| Age | | |
| 25~35 | 139 | 37 |
| 36~45 | 130 | 34.7 |
| 46~55 | 80 | 21.4 |
| 56 above | 27 | 6.9 |
| Job | | |
| Agriculture/Forestry/Fishery | 1 | 3 |
| Civil service | 18 | 4.8 |
| Teacher/Academy lecturer | 9 | 2.4 |
| Professional | 26 | 6.9 |
| Executive position | 7 | 1.9 |
| Office worker | 177 | 47.1 |
| Production/labor | 19 | 5.1 |
| Service/sales | 13 | 3.5 |
| Self-employed | 19 | 5.1 |
| Freelancer | 12 | 3.2 |
| Housewife | 47 | 12.5 |
| Student | 14 | 3.7 |
| Inoccupation | 7 | 1.9 |
| Other | 7 | 1.9 |
| Monthly visit frequency | | |
| 4~9 | 227 | 60.4 |
| 10~15 | 106 | 28.1 |
| 16~21 | 29 | 7.7 |
| 22~27 | 4 | 1.1 |
| 28~30 | 10 | 2.7 |
| Length of time using SNS (year) | | |
| 1~2 | 41 | 10.9 |
| 2~3 | 45 | 12.0 |
| 3~4 | 55 | 14.6 |
| 4~5 | 196 | 52.1 |
| Other | 39 | 10.4 |

We have performed further analyses to see if there are any significant moderating effects of demographic variables (e.g., age, gender, dining frequency). However, we couldn't find any significant effect.

### 5.2. The Measurement Model

Confirmatory factor analysis (CFA) on the entire set of constructs was conducted to test for convergent validity and discriminant validity of the measures. As shown in Table 2, the fit indices are: $\chi^2$ = 404.548 with d.f. = 244 ($\chi^2$/d.f. = 1.658), *p*-value = 0.000; goodness-of-fit-index (GFI) = 0.922; normed fit index (NFI) = 0.945; comparative fit index (CFI) = 0.977;

root mean square error of approximation (RMSEA) = 0.042. These statistics with an exception of the chi-square statistic supported appropriate measurement quality [82]. Almost all standardized factor loadings were found to be above 0.7, indicating evidence of convergent validity [83]. All composite construct reliability (CCR) statistics exceeded 0.7, suggesting reliability of the measures. The average variance extracted (AVE) statistics met the criterion presented by Bagozzi and Yi [83], which is 0.5. The average variance extracted in each measure exceeded the squared correlation estimate between the constructs, suggesting evidence of discriminant validity. Table 3 shows means, standard deviations, and correlations between the constructs.

**Table 3.** Construct inter-correlations, mean and standard deviation.

|  | **1** | **2** | **3** | **4** | **5** | **6** | **7** | **8** |
|---|---|---|---|---|---|---|---|---|
| 1. Authenticity | **0.730** | | | | | | | |
| 2. Consensus | 0.462 | **0.755** | | | | | | |
| 3. Usefulness | 0.635 | 0.532 | **0.771** | | | | | |
| 4. Aesthetics | 0.415 | 0.582 | 0.628 | **0.779** | | | | |
| 5. Cognition-based attitude toward message | 0.593 | 0.558 | 0.645 | 0.547 | **0.879** | | | |
| 6. Affect-based attitude toward message | 0.537 | 0.515 | 0.686 | 0.637 | 0.744 | **0.826** | | |
| 7. Brand attitude | 0.558 | 0.553 | 0.648 | 0.539 | 0.675 | 0.738 | **0.821** | |
| 8. Behavioral intentions | 0.563 | 0.529 | 0.641 | 0.533 | 0.653 | 0.657 | 0.742 | **0.826** |
| Mean | 3.94 | 4.70 | 4.63 | 4.77 | 4.31 | 4.42 | 4.56 | 4.47 |
| SD | 1.05 | 0.77 | 0.98 | 0.96 | 1.09 | 1.10 | 1.01 | 1.01 |

Note: All correlations were significant at the level of $p = 0.01$. Diagonal values in bold correspond to the square root of AVE. Off-diagonal entries are the correlations between the latent variables.

Normality was checked using values of kurtosis and skewness. As shown in Table 2, normality was not a problem because the values of kurtosis and skewness were less than |2.0| and |9.0|, respectively [84].

### 5.3. The Structural Model and Testing of the Hypotheses

We used AMOS 22.0 to analyze the structural model. The overall model fit was satisfactory: $\chi^2 = 425.484$ with d.f. = 252; $\chi^2/\text{df} = 1.688$, $p$-value = 0.000; GFI = 0.919; NFI = 0.942; CFI = 0.976; RMSEA = 0.043. Squared multiple correlation (SMC; $R^2$) statistics for all structural equations were relatively high, all exceeding 70%. Table 4 and Figure 1 show the results of Structural Equation Modeling (SEM) with standardized path coefficients.

**Table 4.** Standardized structural estimates.

|  | **Paths** | **Coefficients** | **$t$-Value** | **$p$** | **Result** |
|---|---|---|---|---|---|
| H1-1 | Authenticity → Cognition-based attitude | 0.289 | 4.080 | 0.000 ** | Supported |
| H1-2 | Authenticity → Affect-based attitude | 0.122 | 1.680 | 0.093 n.s. | Not supported |
| H2-1 | Consensus → Cognition-based attitude | 0.253 | 3.346 | 0.000 *** | Supported |
| H2-2 | Consensus → Affect-based attitude | 0.030 | 0.393 | 0.694 n.s. | Not supported |
| H3-1 | Usefulness → Cognition-based attitude | 0.420 | 4.407 | 0.000 *** | Supported |
| H3-2 | Usefulness → Affect-based attitude | 0.439 | 4.421 | 0.000 ** | Supported |
| H4-1 | Aesthetics → Cognition-based attitude | 0.043 | 0.534 | 0.593 n.s. | Not supported |
| H4-2 | Aesthetics → Affect-based attitude | 0.316 | 3.750 | 0.000 ** | Supported |
| H5 | Cognition-based attitude → Brand attitude | 0.577 | 4.818 | 0.000 ** | Supported |
| H6 | Affect-based attitude → Brand attitude | 0.313 | 2.710 | 0.007 ** | Supported |
| H7 | Cognition-based attitude → Behavioral intentions | 0.536 | 3.888 | 0.000 ** | Supported |
| H8 | Affect-based attitude → Behavioral intentions | −0.170 | −1.421 | 0.155 n.s. | Not supported |
| H9 | Brand attitude → Behavioral intentions | 0.512 | 5.130 | 0.000 ** | Supported |

**Table 4.** *Cont.*

| Paths | Coefficients | *t*-Value | *p* | Result |
|---|---|---|---|---|
| *SMC* ($R^2$) | | | | |
| Cognition-based attitude toward message | | | 0.797 (79.7%) | |
| Affect-based attitude toward message | | | 0.671 (67.1%) | |
| Brand attitude | | | 0.754 (75.4%) | |
| Behavioral intentions | | | 0.742 (74.2%) | |
| *Fit indices* | | | | |
| $\chi^2$ | | | 425.484 | |
| df | | | 252 | |
| $\chi^2$/df | | | 1.688 | |
| *p* | | | 0.000 | |

** $p < 0.01$, *** $p < 0.05$. GFI = 0.919, NFI = 0.942, CFI = 0.976, RMSEA = 0.043. [n.s.] = not significant.

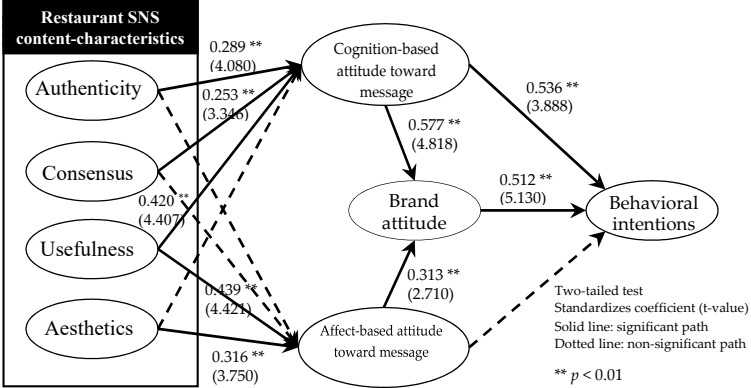

**Figure 1.** Estimates of the structural model.

H1 through H4 were concerned with the effect of social media content characteristics on attitude toward the message. H1 addressed the effect of authenticity on consumer's cognition-based attitude and affect-based attitude. The result shows that authenticity has a significant effect on cognition-based attitude but no effect on affect-based attitude. H2 was related with the effect of consensus. The study finds that consensus has a significant effect on cognition-based attitude but no effect on affect-based attitude. H3 predicted the effect of usefulness on cognition-based attitude and affect-based attitude. The result shows that usefulness is positively related to both cognition-based and affect-based attitudes, supporting H3. H4 addressed the effect of aesthetics. The finding shows that aesthetics has a significant influence on affect-based attitude but no effect on cognition-based attitude.

H5 and H6 were concerned with the impact of cognition-based attitude (H5) and affect-based attitude (H6) toward the message on brand attitude. These hypotheses were supported. H7 and H8 addressed the effect of cognition-based attitude (H7) and affect-based attitude (H8) toward the message on behavioral intentions. The study finds that only cognition-based attitude has a significant direct effect on behavioral intentions, supporting H7 and rejecting H8. As expected, brand attitude is found to have a significant effect on behavioral intentions, thus supporting H9.

## 6. Discussion and Implications

### 6.1. Theoretical Implications

Our study finds that authenticity and usefulness have a significant effect on cognition-based attitude. Authenticity and usefulness of the message are content-related properties and their evaluation requires a high level of cognitive processing, thus showing their positive relationship with cognitive attitude. This finding is consistent with prior research [68] that suggests messages requiring a high level of elaboration (e.g., authenticity, usefulness)

lead to a high level of cognitive responses. Our study also shows that aesthetics plays a significant role in affecting affect-based attitude. This result is understandable as messages using a peripheral cue (e.g., aesthetics) will require a low level of cognitive response and evoke a high level of affective response. For example, music as a peripheral cue in a retail environment is used successfully to ameliorate consumer's perceptions of wait time and to arouse positive emotion [85]. While authenticity, consensus, and aesthetics have an influence on either cognition-based or affect-based attitude, usefulness is found to have an influence on both cognition-based and affect-based attitude.

The finding that usefulness affects both cognition-based and affect-based attitude is interesting because it indicates that restaurant patrons are involved affectively even when they are in the cognitive processing of evaluating usefulness of the information. Some studies [12,13,27] suggest that cognitive processing does not occur in a vacuum of affect. For example, Morris et al. [13] reveal that cognitive elaborators show a higher level of affective responses than cognitive misers [13]. The interplay between cognitive and affective processing is found in some other previous studies [13,29,38]. Ha et al. [38] show that perceived informativeness is related with perceptions of entertainment, suggesting an interaction between cognitive and affective responses. Anand et al. [27] based on their experimental study suggest that affective responses are the last step of a series of cognitive processes. Our study supports the notion that cognitive and affective spheres coexist in forming an attitude. A theoretical implication is that studies based on the ELM or attitude formation should consider affect in the route to persuasion as affect may appear in the cognitive sphere of information processing. Another explanation for the significant impact of usefulness may be found in the cost-benefit paradigm [37]. According to the cost-benefit paradigm, people make a decision based on the comparison between efforts required (cost) and likely results (benefits). Usefulness is an element that increases perceived benefits as it satisfies consumer's information need and is directly related to purchasing task performance. Thus, consumers are more likely to accept the influence when the information is useful.

Our study reveals that cognition-based attitude toward the message exerts a significant direct impact on behavioral intentions, in addition to its indirect effect through brand attitude. Although behavioral intentions are primarily driven by brand attitude, the direct effect of cognition-based attitude on behavioral intentions is significant. While brand attitude partially mediates the relationship between cognition-based attitude and behavioral intentions, it plays a full mediating role in the relationship between affect-based attitude and behavioral intentions. A finding based on our post-hoc analysis involving the Sobel test, gives an insight into three different paths, through which consumers accept an influence. The first path involves a direct link between cognition-based attitude and behavioral intentions. The second path is concerned with brand attitude playing a partial mediating role in the relationship between cognition-based attitude and behavioral intentions. The third path involves brand attitude fully mediating the impact of affect-based attitude on behavioral intentions. The full mediating effect means that the relationship between affect-based attitude and behavioral intentions is fully explained by brand attitude. The finding illustrates the important role of brand attitude in transforming attitude toward the message to behavioral intentions.

Lastly, our study adds evidence to the literature that attitude is multidimensional (cognitive and affective) and a hierarchical response model (i.e., attitude toward message—attitude toward brand—behavioral intentions) is useful for understanding consumer behavior. More importantly, our study, by linking social media content characteristics to attitude formation sheds light on the role of social media marketing communication in influencing restaurant patrons' behavior.

*6.2. Managerial Implications*

Our study shows that both content-quality-related characteristics and peripheral cues exert a considerable influence on attitude formation. Thus, a combination of central and

peripheral routes of persuasion should be used for changing attitudes of restaurant patrons. More specifically, we offer the following suggestions. Our study finding points to the important role of usefulness. Usefulness is found to be the only variable that affects both cognition-based attitude and affect-based attitude and accounts for the largest amount of variances. One plausible explanation is that restaurant patrons may place a utilitarian need before a hedonic need in searching for information. Useful information that allows consumers to reduce time, effort, and perceived risk associated with information search seems to be the most important criterion in evaluating social media content. Restaurant firms should make usefulness a strategic priority by providing an adequate platform in which consumers can post and exchange useful information. For example, a restaurant company may want to offer a platform where consumers can exchange helpful ideas and suggestions (e.g., recommending a certain dish at a restaurant). In an effort to boost consumer's participation in sharing useful information, restaurant firms may want to reward those whose ideas and postings receive favorable responses. In addition, simple feedback tools such as a "useful?" button may be used on the social media sites to gather consumer feedback.

Authenticity is positively related to cognition-based attitude. The finding suggests that a message considered inauthentic will have a negative effect on attitude. Thus, restaurant firms should make efforts to prevent untrustworthy information from influencing consumers. Some restaurant firms offer rewards and incentives to encourage consumers to post food reviews and make a recommendation. This action may result in a multitude of inflated and inauthentic reviews. It may be necessary for restaurant firms to have a communication policy so that any reward or incentive received is revealed.

Our study shows that aesthetics is positively related to affect-based attitude. Aesthetics matters as consumers are typically drawn to a visually attractive object and form a favorable attitude. The finding suggests that restaurant firms should pay attention to not only content quality or message arguments but also physical attributes of the site such as visual effects. Aesthetics may be more important when restaurant firms are trying to create an emotional connection with consumers. For example, in the restaurant industry, a nice visual presentation of food and drink will be helpful for making consumers emotionally connected to the brand.

Our finding that brand attitude plays a full mediating role between affect-based attitude and behavioral intentions deserves a recognition. The finding suggests that brand attitude plays an important mediating role in transforming affect-based attitude to behavioral intentions. Restaurant firms need to make sure that any tactics (e.g., visual effects) used for creating an emotional bond with consumers are consistent with the brand itself, in order to link affect-based attitude to brand attitude. For example, out of many visual images available, restaurant firms may want to choose an image that is directly related to enhancing and supporting the brand image (e.g., restaurant food) as opposed to an image unrelated to the brand (e.g., scenery of the neighboring area).

### 6.3. Limitations and Future Research

Our study has several limitations. Although our study considered four important characteristics of social media content and their impacts on attitude formation, some other characteristics (e.g., background color) may be examined in future research. In doing so, future studies may want to include more variables related to content quality and heuristic cues and compare their effects on attitude formation. Second, our study used a survey method for data collection. The study could have achieved more robust results if it employed an experimental design, in which content quality related characteristics and heuristic cues were manipulated. Third, our study was conducted in South Korea, where collectivist culture is prevalent. One of the variables we examined was consensus. This variable may have a different role in societies characterized of individualism. For example, Bei et al. [2] who examined the effect of online information on consumers in U.S. versus Taiwan showed that Taiwanese consumers considered online sources of information to be more important

than their counterparts. It will be interesting to see how consensus functions in individualist cultures. Lastly, based on the ELM, we assumed authenticity and usefulness would require a higher level of elaboration (a central route of processing), while consensus and aesthetics would require a lower level of elaboration (a peripheral route of processing). Further research is necessary to confirm that consumers actually engage in a higher level of elaboration for authenticity and usefulness and a lower level of elaboration for consensus and aesthetics.

**Author Contributions:** All authors contributed to the content of this paper. All authors have read and agreed to the published version of the manuscript.

**Funding:** This research received no specific grant from any funding agency in the public, commercial, or not-for-profit sectors.

**Institutional Review Board Statement:** Not applicable.

**Informed Consent Statement:** Not applicable.

**Conflicts of Interest:** The authors declare no conflict of interest.

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
