# Peer review of "Characteristics of Social Media Content and Their Effects on Restaurant Patrons"

_sustainability, doi:10.3390/su13020907_

Round 1
Reviewer 1 Report
I think this paper is interesting and well written. Minor revisions are necessary for the publication decision.
I am not sure how the literature in section 2.2. is related to this study.
In section 2.3., it is necessary to add more explanation of why the four characteristics were selected (at least the authors need to provide references). In the discussion of usefulness, ease of use is mentioned, but it is not directly related to the study.
On page 9, please check the appropriateness of checking the discriminant validity.
On page 9, the authors stated that all standardized factor loadings were above .70. However, two standard factor loadings were less below .70. Please check.
On page 9, related to the discriminant validity, the authors stated that “The average variance extracted in each measure exceeded the respective correlation estimate between the constructs.” The expression is not accurate; it should be squared correlation.
On page 13, line 469. The interpretation of the full mediating role of brand attitude between affect-based attitude toward message and behavioral intentions sounds inappropriate. Although all steps are not provided in the paper, it is clear that the relationship between affect-based attitude toward message and behavioral intention is significant. When affect-based attitude toward message is estimated along with the brand attitude, the relationship between affect-based attitude toward message and behavioral intentions becomes non-significant. This clearly demonstrates that the authors claim that “the impact of affect-based attitude on behavioral intention is not likely without the presence of brand attitude,” is not appropriate. Instead, the finding illustrates that the relationship between affect-based attitude toward message and behavioral intention is fully explained by brand attitude. I think more emphasis has to be placed on the indirect effect of affect-based attitude toward message on behavioral intentions through brand attitude. Further, such a finding does not necessarily support the claim “cognitive and affective responses have their own distinctive roles.”
On Table 2, the reported CRs and AVEs are not accurate according to the given standard factor loadings. Please check for accuracy (Not sure if these errors are caused by the misreporting of the standardized factor loadings).
For the model fit report, the authors concluded that the overall model fit was satisfactory. It is true for all but not for the Chi-square test. P-value needs to be rewritten to p < .001 for all p-value of .000.
In Table 4, of fix indices, Chi-square is provided in the table, but others are reported below the table. Fit indices need to be provided together.
Editorial suggestions.
I think this paper needs help from a professional editor. On the very first page in the abstract, the purpose is not properly stated (on the second page as well).
In Figure 1, in the restaurant SNS content characteristic box, change ‘characteristic’ to ‘characteristics’ as there are four characteristics. Adjust space so that ‘authenticity’ is fully shown.
In the Table2, capitalize the first letter of words in the sentences.
Author Response
Thank you very much for giving us the opportunity to revise and improve our manuscript. We have made changes to the paper based on your comments to us. They are listed below and are matched to the points made in your review.
Response to Reviewer #1
Reviewer: I think this paper is interesting and well written. Minor revisions are necessary for the publication decision. I am not sure how the literature in section 2.2. is related to this study.
Our response: Thank you for your insightful comment related to section 2.2. We included this section under Theoretical Frameworks to lay out a foundation for arguing for the role of consensus later on. We are happy to remove this section if you prefer.
Reviewer: In section 2.3., it is necessary to add more explanation of why the four characteristics were selected (at least the authors need to provide references). In the discussion of usefulness, ease of use is mentioned, but it is not directly related to the study.
Our response: Thank you for your insightful comment related to section 2.3. We changed this section into the following.
2.3. Characteristics of Social Media Message Content
Prior research [3,34-40] examines different characteristics of marketing communication content including vividness, usefulness, novelty, neutrality, reliability, authenticity, and consistency. Our study focuses on four major content characteristics that are pertinent to the restaurant industry. Restaurant customers tend to rely on others’ opinions when gathering information and choosing a particular restaurant [42]. Prior research suggests that information should be authentic [43], useful [59, 60], of consensus [10], and be presented aesthetically [12, 63] in order to be effective.
We mentioned ‘ease of use’ only once and very briefly in explaining the TAM. We did not want to omit ‘ease of use’ when discussing the TAM. If you want us to remove the discussion related to ‘ease of use’, we are happy to do so.
Reviewer: On page 9, please check the appropriateness of checking the discriminant validity. On page 9, the authors stated that all standardized factor loadings were above .70. However, two standard factor loadings were less below .70. Please check. On page 9, related to the discriminant validity, the authors stated that “The average variance extracted in each measure exceeded the respective correlation estimate between the constructs.” The expression is not accurate; it should be squared correlation.
Our response: Thank you for being thorough. We corrected the term “correlation” to “squared correlation.” We also changed “All standardized factor loadings” to “Almost all standardized factor loadings.”
Reviewer: On page 13, line 469. The interpretation of the full mediating role of brand attitude between affect-based attitude toward message and behavioral intentions sounds inappropriate. Although all steps are not provided in the paper, it is clear that the relationship between affect-based attitude toward message and behavioral intention is significant. When affect-based attitude toward message is estimated along with the brand attitude, the relationship between affect-based attitude toward message and behavioral intentions becomes non-significant. This clearly demonstrates that the authors claim that “the impact of affect-based attitude on behavioral intention is not likely without the presence of brand attitude,” is not appropriate. Instead, the finding illustrates that the relationship between affect-based attitude toward message and behavioral intention is fully explained by brand attitude. I think more emphasis has to be placed on the indirect effect of affect-based attitude toward message on behavioral intentions through brand attitude. Further, such a finding does not necessarily support the claim “cognitive and affective responses have their own distinctive roles.”
Our response: Thank you for your helpful suggestion. We have removed our inaccurate interpretation of the full mediating effect of brand attitude on the relationship between affect-based attitude and behavioral intentions. The revision is copied and pasted below for your convenience.
“The full mediating effect means that the relationship between affect-based attitude and behavioral intentions is fully explained by brand attitude. The finding illustrates the important role of brand attitude in transforming attitude toward the message to behavioral intentions.”
Reviewer: On Table 2, the reported CRs and AVEs are not accurate according to the given standard factor loadings. Please check for accuracy (Not sure if these errors are caused by the misreporting of the standardized factor loadings).For the model fit report, the authors concluded that the overall model fit was satisfactory. It is true for all but not for the Chi-square test. P-value needs to be rewritten to p < .001 for all p-value of .000. In Table 4, of fix indices, Chi-square is provided in the table, but others are reported below the table. Fit indices need to be provided together.
Our response: Thank you for your comments. We checked all values, but found the reported CRs and AVEs are accurate according to the given standard factor loadings.
Reviewer: Editorial suggestions. I think this paper needs help from a professional editor. On the very first page in the abstract, the purpose is not properly stated (on the second page as well). In Figure 1, in the restaurant SNS content characteristic box, change ‘characteristic’ to ‘characteristics’ as there are four characteristics. Adjust space so that ‘authenticity’ is fully shown. In the Table2, capitalize the first letter of words in the sentences.
Our response: Thank you for your suggestion. We revised the sentence related to the study purpose. For your convenience, we have copied and pasted the revisions below.
“The purpose of this study is to examine four characteristics of social media content and their effects on restaurant patrons.”
“….the purpose of the current study is to identify characteristics of social media content that are relevant to restaurant patrons.”
We made some revisions as recommended by you. We changed ‘characteristic’ to ‘characteristics” in Figure 1. Also, we adjusted space to have ‘authenticity’ fully shown. For Table 2, we capitalized the first letter of the words.
Reviewer 2 Report
Dear Authors:
I congratulate all of you for this paper. I find it provides valuables contributions to the field.
I appreciate the quality of the theoretical foundations and of the statistical description in this paper. The only thing I want to point out is that some assumptions on which some statistical procedures are carried out are not mentioned and that if they are not fulfilled they can give rise to certain errors (e.g. assumption of normality of the data in the Sobel test).
Best wishes
MC
Author Response
Thank you very much for giving us the opportunity to revise and improve our manuscript. We have made changes to the paper based on your comments to us. They are listed below and are matched to the points made in your review.
Response to Reviewer #2
Reviewer: Dear Authors: I congratulate all of you for this paper. I find it provides valuable contributions to the field. I appreciate the quality of the theoretical foundations and of the statistical description in this paper. The only thing I want to point out is that some assumptions on which some statistical procedures are carried out are not mentioned and that if they are not fulfilled they can give rise to certain errors (e.g. assumption of normality of the data in the Sobel test).
Our response: Thank you for the kind remark. We appreciate you for reviewing our paper.
Based on your helpful comment, we have included the following sentences to address normality.
Normality was checked using values of kurtosis and skewness. As shown in Table 2, normality was not a problem because the values of kurtosis and skewness were less than |2.0| and |9.0|, respectively (Schmider, Ziegler, Danay, Beyer, & Bühner, 2010).
Schmider, E., Ziegler, M., Danay, E., Beyer, L., & Bühner, M. (2010). Is it really robust? Reinvestigating the robustness of ANOVA against violations of the normal distribution assumption. European Journal of Research Methods for the Behavioral and Social Sciences, 6, 147-151.
Reviewer 3 Report
Dear Author(s),
Thank you for submitting the paper and sharing the results of your research.
First of all, I would like to underline that your paper is of satisfying scientific standard: reliable and precise. Secondly, its another advantage is the research area focusing on the marketing aspects of using social media by the restaurant owners; therefore, it contributes to filling in the existing academic gaps. Thirdly, the paper is of interdisciplinary nature: it combines many different research threads, like management, marketing, psychology, and communication/media studies (in the context of social media). It is also worth adding that paper refers to current problems regarding the effectiveness of social media marketing (so it can be valuable for practitioners who look for the advice how to plan and realise efficient social media strategies). The text is well written (except some minor flaws which can be easily improved, i.e. line no. 427), based on clarified argumentation with respectively structured narration and well-grounded theoretical framework which properly presents inspirations for your research.
However, the quality of your paper could be improved by implementing some revisions.
First of all, I would like to know why you decided to study social media in the restaurant business – what was your motivation, how could you explain and argue it? What was so special in this area that inspired you to start the research?
The “Introduction” part is a little bit chaotic: you mix the general approach to social media with a detailed focus on social media usage in the restaurant business. To my mind, this part of the paper would be more legible if you first refer to social media characteristics and after that, move to the issues regarding social media in restaurants.
In line 36, you state that “the role of social media is more notable in the service industry”. I would disagree here, and I find your explanation insufficient. Could add more arguments here, including the proper literature which could stand for your statement?
In the “Introduction” there are some generalizations (line 49). You write: “credibility has something to do with the (…)” – it would be more understandable if you could develop and specify your thought here.
In line 51 you mention that social media are “the most influential advertising and promotion tool” – according to marketing-mix, advertising is a part of a promotion (or communication), so it would be great if you could correct this sentence.
The “Theoretical Framework” is not a typical literature review, which has some advantages and disadvantages. On the one hand, it clearly proves that your research is rooted in well-researched theories, which inspired you and became the fundament for your study.
However, on the other hand, not every subchapter there refers to/mentions the restaurant business. It seems that somehow in several parts (i.e. 2.2, 2.3.1, 2.3.2) you have focused on the general threads and omitted their occurrence within the restaurant industry (for example you have written about the meaning of consensus in general but not in the context of a restaurant, but you have included the restaurant context while referring to usefulness). The “Theoretical Framework” could be supplemented with the “restaurants aspect” in each of the subsections, also to link the content of your paper more closely with its title.
It would also be great to consider adding another subsection with a summary of the literature review on social media use by the restaurant industry, emphasising research gaps resulting from this analysis. Thanks to this, defining your study's theoretical contribution to the adopted research area would be more closely related to the paper’s title.
In lines 190-191, you mention scholars and practitioners who supported you with the advice. I am afraid that this statement is too foggy and general – it encourages to ask you about the details. How many scholars and practitioners have you met with? Who were there? What did they say? Have you interviewed them – and if so, why are these interviews not included in the further parts of the paper?
In the “Hypothesis” part, you do not refer to the restaurant industry at all – it seems as if you focused on general relations between brands, social media, attitudes, etc. In my opinion, the paper’s content should be more bonded with the title and the business sector you have chosen to study.
I also have some questions regarding the “Methodology”. I would like to ask you whether the research was representative (it seems it was not, but it should be clarified).
Also, I have a request for explaining the criteria for choosing the research sample (the way of dining in the restaurants, the time of using the social media, the age of respondents): are there any special/well-argued reasons for that?
I haven’t found information on the data collection period (when, how long).
I would also like to ask you for formulating the research questions for your study.
In the “Result” and “Discussion” parts I would like to learn a little more about the responses given by the respondents: for example, did their demographic characteristics (e.g. age, gender) or the frequency of going to restaurants in any way differentiated the answers given? It would really enrich your paper and – again – bond it with the restaurant industry. The “Theoretical Implications” you could refer to the previously identified academic gaps and clearly indicate your contribution here (in the area of literature referring to social media in the restaurant industry).
Dear Author(s), as I have mentioned – these revisions are of a minor character. I hope you will decide to consider them while improving your paper's quality to enhance the chance for publication of your work.
All the best in 2021.
Sincerely.
Author Response
Thank you very much for giving us the opportunity to revise and improve our manuscript. We have made changes to the paper based on your comments to us. They are listed below and are matched to the points made in your review.
Response to Reviewer #3
Reviewer: Dear Author(s), Thank you for submitting the paper and sharing the results of your research. First of all, I would like to underline that your paper is of satisfying scientific standard: reliable and precise. Secondly, its another advantage is the research area focusing on the marketing aspects of using social media by the restaurant owners; therefore, it contributes to filling in the existing academic gaps. Thirdly, the paper is of interdisciplinary nature: it combines many different research threads, like management, marketing, psychology, and communication/media studies (in the context of social media). It is also worth adding that paper refers to current problems regarding the effectiveness of social media marketing (so it can be valuable for practitioners who look for the advice how to plan and realise efficient social media strategies). The text is well written (except some minor flaws which can be easily improved, i.e. line no. 427), based on clarified argumentation with respectively structured narration and well-grounded theoretical framework which properly presents inspirations for your research. However, the quality of your paper could be improved by implementing some revisions.
First of all, I would like to know why you decided to study social media in the restaurant business – what was your motivation, how could you explain and argue it? What was so special in this area that inspired you to start the research? The “Introduction” part is a little bit chaotic: you mix the general approach to social media with a detailed focus on social media usage in the restaurant business. To my mind, this part of the paper would be more legible if you first refer to social media characteristics and after that, move to the issues regarding social media in restaurants.
Our response: Thank you for your helpful suggestion. We have revised and reorganized the introduction based on your suggestion. The revised introduction now offers a discussion on the general characteristics of social media before making references to the restaurant industry. In order to stress the study’s motivation, we have added the following sentence.
“Studying the role of social media in the restaurant industry is worthwhile because restaurant patrons use social media as an essential tool for gathering information and making a purchase decision.”
Reviewer: In line 36, you state that “the role of social media is more notable in the service industry”. I would disagree here, and I find your explanation insufficient. Could add more arguments here, including the proper literature which could stand for your statement?
Our response: Thank you for your insightful comment. We have added the following sentence in replacement of the sentence above. For your convenience, we have copied and pasted the revision below.
“Studying the role of social media in the restaurant industry is worthwhile because restaurant patrons heavily rely on social media as an essential tool for gathering information and making a purchase decision.”
A citation that supports our argument above was offered a few sentences earlier (i.e., “Social media is the most influential marketing communication tool…..). (https://www.modernrestaurantmanagement.com/10-social-media-marketing-tips-for-restaurants/).
Reviewer: In the “Introduction” there are some generalizations (line 49). You write: “credibility has something to do with the (…)” – it would be more understandable if you could develop and specify your thought here.
Our response: Thank you for pointing this out to us. Credibility is defined as believability of the information source. In order to clarify our point of view, we have added the following sentence.
“For example, consumers tend to believe information posted by fellow consumers more than an advertising message claimed by the company.”
Reviewer: In line 51 you mention that social media are “the most influential advertising and promotion tool” – according to marketing-mix, advertising is a part of a promotion (or communication), so it would be great if you could correct this sentence.
Our response: Thank you for being thorough. You are correct. We have changed the term “advertising and promotion tool” to “marketing communication tool.”
Reviewer: The “Theoretical Framework” is not a typical literature review, which has some advantages and disadvantages. On the one hand, it clearly proves that your research is rooted in well-researched theories, which inspired you and became the fundament for your study. However, on the other hand, not every subchapter there refers to/mentions the restaurant business. It seems that somehow in several parts (i.e. 2.2, 2.3.1, 2.3.2) you have focused on the general threads and omitted their occurrence within the restaurant industry (for example you have written about the meaning of consensus in general but not in the context of a restaurant, but you have included the restaurant context while referring to usefulness). The “Theoretical Framework” could be supplemented with the “restaurants aspect” in each of the subsections, also to link the content of your paper more closely with its title. It would also be great to consider adding another subsection with a summary of the literature review on social media use by the restaurant industry, emphasizing research gaps resulting from this analysis. Thanks to this, defining your study's theoretical contribution to the adopted research area would be more closely related to the paper’s title.
In the “Hypothesis” part, you do not refer to the restaurant industry at all – it seems as if you focused on general relations between brands, social media, attitudes, etc. In my opinion, the paper’s content should be more bonded with the title and the business sector you have chosen to study.
Our response: Thank you for your insightful suggestion. As we alluded in the introduction of the manuscript, to our best knowledge, there is no study that examined social media content characteristics in the restaurant industry. We have conducted another search on ABI/INFORM database using key words such as restaurant industry, social media, content, and attitude. Unfortunately, we couldn’t find any food-service industry study related to social media content. In order to be better aligned with the title of the paper, we have changed the word “consumers” to “restaurant patrons” whenever appropriate. If you like us to remove the word “restaurant” from the title of the paper, we are happy to do so.
Reviewer: I also have some questions regarding the “Methodology”. I would like to ask you whether the research was representative (it seems it was not, but it should be clarified). Also, I have a request for explaining the criteria for choosing the research sample (the way of dining in the restaurants, the time of using the social media, the age of respondents): are there any special/well-argued reasons for that? I haven’t found information on the data collection period (when, how long).
Our response: Thank you for your constructive comment. The sampling criteria were chosen based on our research experience and not necessarily based on the literature (We couldn’t find any related to sampling criteria). We felt that those who have experiences, whether they are about social media networks or restaurants, are better qualified to evaluate their experiences than those who do not. We have added the data collection period (i.e., March 2018) to the methodology section. We have also specified that we used a convenience sampling method.
Reviewer: I would also like to ask you for formulating the research questions for your study.
Our response: We have already included research objectives in the introduction. Research objectives are expressed as statements and research questions are expressed in a question format regarding what we want to study. If you want us to add research questions in addition to the research objectives, we are happy to do so.
Reviewer: In the “Result” and “Discussion” parts I would like to learn a little more about the responses given by the respondents: for example, did their demographic characteristics (e.g. age, gender) or the frequency of going to restaurants in any way differentiated the answers given? It would really enrich your paper and – again – bond it with the restaurant industry. The “Theoretical Implications” you could refer to the previously identified academic gaps and clearly indicate your contribution here (in the area of literature referring to social media in the restaurant industry).
Our response: Thank you so much for constructive feedback. We have performed further analyses to see if there are any significant moderating effects of demographic variables (e.g., age, gender, dining frequency). Unfortunately, we couldn’t find any significant effect. So, we include the following sentence at the end of 5.1 Demographic profile section.
We have performed further analyses to see if there are any significant moderating effects of demographic variables (e.g., age, gender, dining frequency). However, we couldn’t find any significant effect.